# City as a Growth Platform: Responses of the Cities of Helsinki Metropolitan Area to Global Digital Economy

Ari-Veikko Anttiroiko [1,*] , Markus Laine [1] and Henrik Lönnqvist [2]

[1]   Faculty of Management and Business, Tampere University, 33100 Tampere, Finland; markus.laine@tuni.fi
[2]   City Strategy and Management, City of Vantaa, 01300 Vantaa, Finland; henrik.lonnqvist@vantaa.fi
[*]   Correspondence: ari-veikko.anttiroiko@tuni.fi; Tel.: +358-(0)40-190-4017

**Abstract:** The aim of this article is to shed light on how ongoing structural change towards the global digital economy condition urban economic development. Discussion starts with a brief reference to the growth machine thesis and its emphasis on the interests of local land and real estate owners. This theory serves as a contrasting point for the second element of our framework, the platform economy, which brings digital platforms and the transnational capitalist class into the picture. The transition from the urban growth machines of the industrial age to the digital growth platforms of the information age imply a radical change in the context of urban economic development. On this basis, we discuss cities' need to adjust their growth strategies to the conditions of the emerging platform economy. Our illustrative case is the capital region of Finland. We interviewed officials and experts who hold key positions in the design of economic development policy in the three largest cities of this area. The empirical results show that the platform economy is rather vaguely conceptualized, and its challenges are ambiguously addressed. Cities have, however, started to adopt platform and ecosystem thinking in their strategies and established urban innovation platforms, talent and start-up attraction programs, and open data initiatives that reflect the gradual adoption of platform logic in urban economic development.

**Keywords:** platform; platformization; platform economy; platform urbanism; digital economy; economic growth; growth machine; city government; Helsinki Metropolitan Area; Finland

## 1. Introduction

The objective of this article is to shed light on the ongoing structural transformation towards the global digital economy and its impact on the premises of urban economic development. Our hypothesis is that digitalization is changing the preconditions for strategies designed to promote economic growth in cities. To put this change into perspective, we found it instructive to start with Logan and Molotch's [1] growth machine thesis, which was created to describe the development logic of the industrial era with a special view to the role of ownership of land and real estate as the major driver behind the interests of the local growth coalition. Such a location-specificity and anchorage on parochial capital has been weakening for some time, first along with postindustrial tendencies in the postwar decades and more recently with the new economy associated with the economic impacts of digital platforms, communication services, and global networks [2]. Our hypothesis is that as there are some evident changes taking place in the land nexus as an urban phenomenon, it must have some significant implications for the design of urban growth strategies.

A paradigmatic actor in the new economy is a platform company, such as Google, Apple, Microsoft, Facebook, Amazon, eBay, and the like. As important as they have become, this phenomenon cannot be

reduced to them. Rather, we are talking about a profound transformation with a range of manifestations in politics, economy, and social life. In the economy, the issue is essentially about the impacts of digitalization or digital transformation, epitomized by platform businesses in which positive network effects are the source of value creation and competitive advantage. Other novel trends in the economy relate to various dimensions of sharing economy (Lyft, Airbnb, Uber), attention economy that operates on digital platforms (e.g., YouTube celebrities), and superstar effect or superstar economy and related concentration of economic benefits for a handful of successful players in each given domain. All in all, the rise of digital platforms together with closely related economic trends implies dramatic changes in business, work, and social life [2].

John Zysman has called the current transformation in the economy as service transformation, driven by digitalization and networking [3]. Another picture of this new reality has been built by Brynjolfsson and McAfee in The Second Machine Age, which describes how the relationships between productivity, GDP per capita, employment, and median income have been decoupling for some time. This implies that technological progress goes on but its benefits are not distributed as equally as in the industrial era [4]. This development generates almost literally jobless growth, which obviously poses a threat to balanced urban and societal development. The other consequence is the spikiness of the development, which has a tendency to create socially unsustainable winner-take-all urbanism [5].

Some of the current tendencies started to sprout at different pace during the 20th century and the postwar decades, in particular transforming inner logic, industrial structure, and global hierarchy of cities. These kinds of changes were accurately theorized in the 1980s and 1990s, one of the first contributions being John Friedmann's [6] world city hypothesis. He described how world cities manifest themselves in the structure and dynamics of their production sectors and employment. Such a development can be seen to create a two-fold challenge in terms of local responses, which echo the classic dual politics thesis and its ideal-typical dichotomy of politics of production and politics of consumption [7]. In the context of current economic transformation, city governments' promotion of local production and investment functions are challenged by such factors as the decomposition of production, emergence of global production chains and innovation networks, the emergence of platforms, and the increased fluidity of the economy. The other side of the equation is the localization of the social consequences of global digital economy, which translates into the question of how cities can take care of the balanced urban development under circumstances in which polarization tendencies prevail, social problems escalate, and urban fiscal crisis gets worse in shrinking cities in particular. These two perspectives are two sides of the same coin. As put by Friedmann, as a flip side of the formation of basing points in the global economy and related concentration and accumulation tendencies there are social polarization tendencies and increasing social costs, which are becoming a heavy burden to both nation-states and cities. He claims that the economic infrastructure desired by transnational capital and the social reproduction supported by local and national elites creates a social cost that puts a heavy burden on the politically weakest and the most disorganized segments of the population and leads eventually to the marginalization of the poor [6]. The issue is whether such views will gain new relevance when the global urbanization, increased global competition between cities, and the polarization tendencies of the new economy intersect.

The dilemma of locality in the increasingly global, digital, and fluid new economy is the focus of our attention. We assert that before the acceleration of technological development and the globalization of the economy in the 1990s and in the following decades, the local economic development was strongly linked with localities and their national contexts [1]. While location-specific aspects are still important, the accumulation of parochial capital is not necessarily the primary force setting the direction for urban economic development, however. Tendencies associated with the new economy seem to have slowly altered the connection between the place and the economy, affected by reduced transportation costs, intensified global communication, increased importance of the knowledge-intensive value chains, and expanding flow-substitution economy, the latter referring to the flow of resources without dependence on any particular location [8]. Implications for local growth coalitions are manifold, one of

them being the disintegration of urban growth machine, the other being the increased complexity of the urban growth agenda. This implies that it is increasingly difficult to understand the causalities of the economic reality and, consequently, to design economic development policies that divert global value flows to a particular locality.

Previous observations have led to our interest in the evolution of urban growth machine, especially against the radical contextual changes caused by digitalization and the globalization of the economy. In short, our objective is to make sense of how the rise of the global digital economy has affected the premises of local economic development. As there is a need to limit the scope of discussion about contextual changes in order to keep it manageable, we will focus on the platform economy. This is because of the economic significance of the platform business as well as the intriguing connection of platform logic with localities, as seen in the emergence of local innovation platforms and other forms of platform urbanism. We will connect this discussion with urban theories, most notably with theorizations of urban growth coalitions as presented in various strands within critical urban political analysis or new urban politics [9]. We ended up with the following set of research questions, which will be discussed in this article both theoretically and with reference to the empirical case of the three major cities of the capital region of Finland.

a.    How the representatives of city government and local development agencies perceive and conceptualize the platform economy?

b.    How have cities conceived local interests and local embeddedness of economic development in the context of the globally-oriented digital platform economy?

c.    How have cities addressed and documented the aspects of the platform economy in their strategies and in their economic development policies?

d.    What are the factual policy measures and initiatives designed to meet challenges associated with the platform economy?

Conducting research on how cities have understood and responded to a particular emerging trend has its challenges in terms of research design and empirical evidence. As we are dealing with a novel phenomenon, we do not confine ourselves only on what our empirical case—the capital region centering around the city of Helsinki—provides but utilize also theoretical discussions and insights gained from previous research. This is to say that due to the novelty of the phenomenon investigated here, the capital region of Finland is not used as a yardstick but rather as an illustration of local responses to the challenges posed by the global digital economy.

This introductory section is followed by the description of the research setting, method, and data. Then we provide theoretical discussion of four key themes that are essential for our analysis. The next section continues with the description of the empirical case and the empirical analysis of the three cities of the capital region of Finland. In the discussion section, we will ponder the relevance of our empirical findings and explain how they relate to theoretical insights on the local-global interplay. Lastly, in the short concluding section, we wrap up the article and highlight a few key findings.

## 2. Methodology

This is a theoretically oriented case study. We discuss the case of the capital region of Finland, centering around the city of Helsinki. As the topic involves a high degree of novelty and the phenomenon itself is rather ambiguous, it has its implications for what kind of empirical evidence we seek. In short, the case is used as an example of how a technologically advanced and innovative metropolitan area in the northern part of Europe has perceived the rise of digital platform economy and designed responses to it. The above point on ambiguity is of vital importance methodologically, as it dictates the role of the case. That is, the case serves primarily to demonstrate how the representatives of city governments deal with ambiguity and how they create policies and actions under such challenging circumstances.

The capital region of Finland is in many ways an illustrative case. We just have to think of the reputation of Helsinki metropolitan area concerning technology, innovation, design, and start-up culture. All these make it plausible to assume that Helsinki with its neighboring cities in the capital region have designed strategies or at least taken into account in a piecemeal way some aspects of the rising platform economy. This is not necessarily the case with less innovation-oriented cities in Europe. The primary requirement for our case selection is thus that the city or metropolitan area must be developed enough in terms of local governance, industrial structure, and technological development in order to provide sufficient degree of relevance and insights to how the representatives of city government perceive this phenomenon and reflect upon their own strategies and actions.

As we are dealing with a novel phenomenon, case study seems an appropriate research strategy. Our choice implies that we focus on the capital region as a whole. A practical reason behind the selection of the case is that authors have a strong preunderstanding of it. The other, and more important, justification is that due to high degree of novelty of the phenomenon under investigation, the case serves as an exemplification that provides a view of the city-level reality. Methodologically, such a case is referred to as an illustrative case [10]. This implies that the issue is not about the factual responses to the platform economy, for the phenomenon itself is vague and the view of the responses of the city depicts actually an aggregate of several separate actions, which may have only a vague relationship with the platform economy. For the same reason we ended up with a single case study instead of a multiple or comparative case study, which all would have been possible, as our case comprises the three municipalities in the capital region. Taking into account the different roles of cities in the regional structure, the novelty of the issue and the uniform national and regional setting of the three cities in question, neither multiple nor comparative case study would have made a methodologically rigorous setting. Both Espoo and Vantaa have developed as an integral part of the capital region centering around the city of Helsinki. Consequently, a single case study seemed a feasible methodological strategy that provides sufficient empirical evidence of how the platform economy is perceived by key institutional actors in a metropolitan area.

The selection of key informants is of great methodological importance. Formally, the Finnish capital region includes four cities, of which three under investigation in this article are significant by population and economic clout, those of the cities of Helsinki, Espoo, and Vantaa. The main criteria for selecting informants is their managerial and expert positions in local economic development units or development agencies, as this guarantees that their views have a relevant connection with the cities' factual strategy formulations and policy measures. Thus, we selected seven informants from the economic development units and relevant expert organizations of all three cities. The following interviews were conducted by the authors from May to August 2018: CEO of Helsinki Business Hub, Helsinki; Head of Competitiveness and International Affairs, and Director of Economic Development of the city of Helsinki; Director of Business Development, and Data Analytics Consultant of the city of Espoo; and Director of Economic Development, and Economic Development Manager, city of Vantaa (for details, see Appendix A).

Interviews were semistructured theme interviews. The structure of the interviews was built on four main themes reflecting the abovementioned research questions: the conceptualization of the platform economy; the perceptions of local interests vis-à-vis globalized digital platform economy; strategies designed to meet the challenges of the platform economy; and factual policy measures that can be associated with the platform economy (see Appendix B). Interviews were recorded and transcribed. The qualitative interview data was analyzed with a focus on identifying nuances of how the given themes were interpreted and conceptualized and how they were operationalized in the given professional context. Some verbatim spoken words from the transcripts are included in empirical analysis as an evidence of the perceptions among informants and of the current state of development in the given metropolitan area. Verbatim quotations were also used to highlight the uncertainties and ambivalent aspects of local economic development when dealing with novel contextual changes.

## 3. Theoretical Framework

### 3.1. Evolving Urban Growth Machine

Growth machine thesis originated in the 1970s and was elaborated in the following decades to explain the underlying capitalist interest behind the development-oriented urban politics and policy [1,11]. It belongs to the family of theories of the political economy of place, or more specifically, of the commodification of place [12]. The major intervention of this theorization was to show how the coalition of land-based elite in the competitive environment utilizes urban politics and policy in its pursuit of the accumulation of capital and wealth. The initial context of this theorization was the postwar urban development in the United States [13]. Harvey Molotch described the core idea of the theory in his seminal article as follows:

> "A city and, more generally, any locality, is conceived as the areal expression of the interests of some land-based elite. Such an elite is seen to profit through the increasing intensification of the land use of the area in which its members hold a common interest. An elite competes with other land-based elites in an effort to have growth-inducing resources invested within its own area as opposed to that of another. Governmental authority, at the local and nonlocal levels, is utilized to assist in achieving this growth at the expense of competing localities. Conditions of community life are largely a consequence of the social, economic, and political forces embodied in this growth machine." [11] (p. 309)

The interests discussed here are thus rooted in land and real estate. Such a parochial capital had in relative terms greater role in preindustrial era. Yet, the true nature of urban growth machine could not emerge in the conditions of the preindustrial society. The situation started to change due to industrialization and related modernization of Western societies. Critical preconditions for the functioning of urban growth machine were a high urban density, diversification of the functions of the city, and internationalization of the economy. In the Western world, such a development led to the emergence of industrial cities with dual role as concentrations of both productive and reproductive functions. This is how the growth machine formation serves and legitimizes the accumulation of capital: (a) everyone benefits from economic growth, which is (b) promoted on behalf of the community by city government and local stakeholders with a practical aim to (c) increase attractiveness of the urban community in terms of investments, which (d) increases demand for and value of land and built environment. This implies that such a growth machine is essentially local or at least its interest in the accumulation of capital is strongly conditioned by location-specific development activities, such as zoning and urban design and the development of infrastructure and built environment [14]. The other point worth highlighting is that attention is paid here to a particular type of actor, property owners, who strive to maximize the rental value of their property by increasing the attractiveness of the area and thus increasing demand for land and real estate, and by intensifying its uses or focusing on those functions that provide highest rents. They are particularly keen on promoting local economic growth, as "their material interests are geographically rooted" [15].

Yet, there is another form of capital, which Logan and Molotch [1] call metropolitan capital, which flows more relentlessly from one place to another wherever new opportunities arise for gaining higher returns on investment. This is an essential part of the capitalist accumulation under global capitalist system. In practice, metropolitan capital refers to foreign direct investments, portfolio investments, or other equity investments as well as to footloose industries for which it is relatively easy to move from one place to another in the purpose of benefitting from favorable conditions for business. There has been a certain degree of fluidity in the economy since the ancient times, but it started to accelerate at the intersection of accumulated wealth and the creation of world system that provided new opportunities especially in international finance and trade. At this point, it had become clear that developers and investors were not necessarily locally oriented. Their interests were increasingly in volatile global

financial markets and foreign direct investments rather than, say, in local zoning decisions [13,16,17]. This all has had an inevitable impact on the interplay between parochial and metropolitan capital.

Soon after the introduction of growth machine thesis, several new trends emerged to counterbalance or challenge the fundamentals of the functioning of the urban growth machines. Among the first ones was the emergence of environmental and social movements that created not only alternative visions but attacked also the very ideological core of the growth machine, that is, the idea of growth itself. Many movements took directly or indirectly an antigrowth view of urban development [13]. Institutions of public governance had largely a dual role in this game, for as democratic institutions they reflected at the same time the aggregated views of the local electorate, leading to a host of institutional arrangements, policies, and services that were not built with growth machine interests on mind. Yet, city governments bear the responsibility of local economic development, having in many cases fairly developmentalist orientation with a clear focus on promoting economic growth and, while doing it, collaborating, and being influenced by local business partners.

The other challenge, which appeared as a natural result of the acceleration of globalization, was the emergence of transnational capitalist class, which challenged the view of local growth coalition's integrity and loosened its connection with locality [18]. This alienated further the strategically oriented growth coalition from the needs of local inhabitants.

Beside these, the third corner stone of the theory, a place or a locality, has also been changing due to both globalization and digitalization and related emergence of networking and platformization as new prevailing principles of social and economic organization, which increased delocalization [19]. This is decisive in judging the validity of growth machine thesis. Namely, digitalization was assumed to make the world "flat", which obviously affects the local-global dialectic in economic life [20]. However, it seems that the world has actually become increasingly spiky [21,22], which matches well with the overall asymmetrization tendency of the global digital economy. Thus, rather than making localities insignificant altogether, the tendencies of global digital economy are increasing global urban asymmetry.

Even if there have been continuous disintegrative and disruptive elements in the formation of urban growth coalitions, growth machine has maintained some of its appeal over time. This is surprising, for practically all fundamentals on which the political economy of place has been anchored experienced a radical change due to the rise of globalization and digitalization in the 1990s and the following decades. Logan and Molotch [1] addressed some of these issues, but the radical global transformation had not yet matured at that time. Overall, it looks as if growth machine discourse remained largely confined with a somewhat narrow and structurally oriented view of urban politics. On the other hand, by the late 1980s many urban theorists started to theorize about the integration of local economies into global economy. It gave an impetus to the rise of New Urban Politics, which explicitly denied the localist or place-centric view of local development by adhering to political and economic forces that condition local development [13].

Our hypothesis is that growth machine literature is not able to provide a sufficient integrative view of the political economy of place, primarily due to its insufficient attention to the local-global dialectic deeply affected by the contextual change. In order to shed light on this issue, let us discuss next the emergence of new economy and one of its most important manifestations, digital platforms, which is used here to illustrate the new locus of growth at the intersection of dispersed sense of place and the global digital economy. What remains "local" in a new growth agenda is theoretically and empirically one of the most fascinating dilemmas of our time. It resonates with broader issue of the relationship between the historically rooted identities of city dwellers and their communities and the global networks of instrumental exchanges [19]. In addition, there is something even more intriguing in this picture. Namely, if a long-term trend in terms of economic value has been a transition towards metropolitan capital, the next phase may be the platformization of the economy, which emerges at the intersection of global networks, technological interconnectedness, and systemic intelligence. Hence

a hypothesis of the gradual transformation from locally-bound growth machine to globally connected urban growth platforms as the locus and model of urban economic growth.

### 3.2. Platform Economy

Technology determines the mode of development, which conditions the current state of development in the economic life through the technological arrangements that determine the level of productivity and indirectly also of surplus. Technological innovations are the key to the evolution of the mode of development, which broadly speaking evolved from agrarian to industrial since the 18th century and started to shift towards informational mode of development during the postwar decades [16,19]. Among the most important underlying factors behind this last shift was increased computing power, which together with global digital networks converted into economic tools using algorithms that operate on the raw material of data. Moreover, these digitized processes have moved to the cloud, where they can be easily accessed, creating the infrastructure on which the entire platform-based markets and ecosystems operate. Platforms and the cloud are an essential precondition for what has been called the "third globalization," being capable of reconfiguring globalization of the economy itself [2]. As one of its instances, global value chains have been reshaped by a global fragmentation of production, increased specialization, and global production and innovation networking. Furthermore, the increased utilization of robotics and artificial intelligence in both production and consumption on a global scale are assumed to have far-reaching impact on business and labor markets [23].

Conceptually, the economy where "business firms have learnt to take advantage of both the ICT revolution and the globalization of business activities in ways which improve productivity", is called new economy [24]. It is close to a broadly defined digital economy [25]. In this sense, the impact of radical innovations, new technologies, and digital networks on productivity, innovativeness, and competitiveness is characteristic to new economy. In terms of industries, it encompasses primarily high-tech industries such as information technology including computer software and hardware as well as Internet-based businesses, such as social media, communication services, and the internet and catalog retail business. However, "new economy" does not tell much about the nature of this shift in economic life. To depict its essence, we may associate it with technology, high-tech industry, internet, digital networks, platforms, sharing, and so forth. In this article, we refer to this new economic landscape as global digital economy. Furthermore, as a more precise term that captures the essence of this economy from the point of view of the high-profile companies and the key mechanisms that facilitate new economy sectors and bring key players together, we refer to it as platform economy. In short, we see platforms as the key integrators and facilitation mechanisms in the global digital economy.

Platforms enable the online mediation of social and economic interactions and transactions. The most powerful or effective platforms that characterize new economy are digital platforms. They include digital marketplaces and consumer discretionary firms such as Amazon, eBay, and Alibaba, special forms of sharing and the matching of demand and supply in service industries as in the cases of Uber and Airbnb and Internet and communication companies such as Google, Baidu, Facebook, and Tencent, which are at the forefront of the new economy. The term "platform" points to a set of online digital arrangements whose algorithms serve to organize and structure economic and social activities. In the IT world, the term means a set of shared technologies and interfaces that are open to a broad set of users who can build their services, applications, or solutions on a stable substrate [2]. Platforms attract users and facilitate and monetize their transactions, relying on decentralized logic of value creation [26–28]. In addition, such a business model is smart in the sense that platforms utilize algorithms and other forms of artificial intelligence in their facilitation processes, which together with huge processing capacity allow exchanges at an unprecedented scale. Such a business model has inherent risks too, which has stirred up discussion about privacy, power, and regulation relating to platform economy [29–31].

### 3.3. Social and Economic Consequences of Platformization

Technological progress is supposed to help everyone by improving their income and various aspects of well-being. Brynjolfsson and McAfee [4] call such benefits "bounty". The other side of the equation is how equally these benefits are distributed in society, which they call "spread". One of their arguments is that the age of bounty is not obviously contributing to everyone's well-being. Empirical data provides some evidence for a claim that people are losing ground over time, even by absolute terms: "In America, the income of the median worker is lower in real dollars than it was in 1999 and the story largely repeats itself when we look at households instead of individual workers, or total wealth instead of annual income. Many people are falling behind as technology races ahead." [4] (p. 168). Another snapshot of the current development is the comparison between wages and living costs. For example, from 1990 to 2008 median family income grew in the US about 20%, while housing and college expenses grew by over 50% and health care by more than 150% [32]. This and a host of similar trends point to polarization tendencies even in the United States, which is a country that has captured a large share of values generated by platform companies.

Brynjolfsson and McAfee [4] (pp. 126–128) have made an interesting comparison with Kodak and Instagram and Facebook, which illustrate both productivity jump and polarization tendencies. In this scene Kodak represents the "first machine age" and Instagram and Facebook the platform economy, or the "second machine age" as Brynjolfsson and McAfee call it. Instagram with its 15 workers created a simple application, which attracted over 130 million users. Facebook bought Instagram for over 1 billion in 2012. Facebook had at that time some 4600 employees, while Kodak, during its peak time in the 1970s, had as many as 140,000 employees, one-third of them in Rochester, New York. This comparison is most telling. The most striking point is the difference between the bounty and spread of wealth in the first and second machine age. Platform companies like Facebook have much more customers and greater market value than Kodak ever had, yet they employ only a fraction of people what Kodak and similar high-tech companies of the industrial era employed. To generalize, for almost two hundred years, wages increased alongside productivity, which provided some legitimation to claim that everyone benefits from technological advancements. Recently, however, median wages have stopped tracking productivity. Such a decoupling is an important indication of the nature of platform economy.

Previous observation hints that the rise of the platform economy affects the relationship between capital and labor. Capital has for a long time substituted for labor with the help of technology. However, the share of GDP going to labor has been fairly stable, until recently. Wages and the standard of living rose almost as dramatically as productivity. In the United States, for example, several decades after the World War II, labor share of GDP was 64.3% on average, while it reached much lower point, 57.8%, in the third quarter of 2010 [33]. It is worth closer inspection whether this will gradually affect practically all advanced economies that are integrated into the global digital economy. Thus, if the productivity grows while labor as a whole is not capturing the value, the next question is, who actually captures that value. Brynjolfsson and McAfee [4] (pp. 143–145) make a legitimate assumption that the answer is capital owners.

### 3.4. Globally Connected Urban Growth Platforms

If the dynamism of economic life changes, such changes will have their reflection in urban life in one way or another. One of the consequences is that the idea of growth is changing as is also the composition of growth coalition. The underlying interests of land and real estate owners continue to exist, but it is not necessarily their interests that dominate the setting of urban growth agenda. The consequence is the gradual emergence of new growth platforms that are not only more diverse, diffuse, and dispersed than what was the urban growth machine in the industrial era but also more genuinely global and innovation and talent oriented.

If the old growth coalition harnessed the entire city in the interest of accumulating parochial capital, how valid is this view in the global digital economy? We can shed light on this issue by

considering not only how digitalization in general is transforming global flows but also the fact that the knowledge-intensive portion of global flows dominates capital- and labor-intensive flows, which is an indication of a paradigm shift in international trade. They account today for about a half of global flows, and the figure is expected to rise steadily in the coming years. This is drastically different from the situation in the industrial era when global flows were dominated by labor-intensive flows from low-cost manufacturing countries and commodity-intensive flows from resource-rich economies. [34]. What is thus the relative importance of locality and related parochial capital in economic life and decision making if the factors of production are increasingly fluid, business processes are digitally facilitated, and knowledge-intensive products and services dominate global flows of values?

On the ground of current developments in the economic life, including the emerging tendencies associated with the new economy, we may have to reconsider the idea of locally oriented growth machine profoundly influenced by local land and real estate owners. In the industrial age, growth machines were organized on a local basis with a reliance on land-use intensification. This scene started to change gradually due to the impact of informational mode of development, even though digitalization, agglomeration, and urban form appeared to have a complex and rather nonlinear relationship [35]. For example, knowledge-intensive clusters at the core of informational economy show agglomeration tendencies and have even localized close to the infrastructural nodes, such as road and rail structures and terminals [36]. However, the supply-chain expansion and the intensification of circulatory possibilities in a regional transportation network seems to have catalyzed a move away from place dependence, resulting in the emergence of multicity growth coalitions [37]. Due to current radical developments in the economy, we have the urge to look even further. We find several reasons to hypothesize that it is the transnational capitalist class that assumes a dominant role in affecting locality development under the conditions of platform economy (cf. [18]). It defines its relationship with localities according to its higher-level goals—as a rule without a predetermined obligation towards a particular locality—within global networks of instrumental exchanges [19]. It is in a position to exercise significant power with regard to global processes with obvious local ramifications, relying on its ownership and control of the means of production sustained by relevant interlocked agencies, which involves not only business circles but bureaucracy and various high-profile professions as well [38]. With respect to industries, focus is less on manufacturing and more on knowledge-intensive and innovation-intensive firms that have their stake in the platform economy. Regarding the latter, there is a natural tendency to rely on creativity and innovativeness, which connects such activities with Richard Florida's thesis on creative class and related talent attraction scheme [39].

Globalization has given impetus to the reorganization of urban space [40], which has direct connection with locality development. One is a desperate need for global attractiveness, which can be promoted via international city branding. This has increased the relevance of wow, star, and iconic architecture [18,41]. The other form of the reorganization of space is the increased role of special zoning, which is applied in order to attract specific global flows and to perform specialized global functions in the global digital economy, such as free trade zones, science parks, financial districts, logistics zones, outsourcing hotspots, thematic urban districts (health care, education, design, arts etc.), and wellness communities, each having their own appeal to international business, investors, talent, visitors, and wealthy residents [42].

### 3.5. City Governments' Adjustment Strategies

Previously outlined economic conditions have an impact on cities as collective entities, including their growth strategies. Our assumption is that cities are gradually learning about how the global economy is changing and consequently apply new models and tools that help them to adjust to global digital economy. Some cities have also built local growth platforms, which are designed to mix the fluidity of the new economy with the city government's ultimate need to channel benefits to their locality or at least to those institutional actors that have a connection with the given locality.

According to Savitch and Kantor, the major approaches to cope with globalization challenge include growth strategies, community development initiatives, regionalism, and national urban policy [43]. Of these the first is the only one that focuses on promoting economic growth and implies a strategic adjustment to global conditions, while the other three focus on rescaling the city to enhance bargaining leverage in the international marketplace. All of them pursue in a slightly different ways the enhancement of local governmental influence over the global capital flows. While national urban policy may have been most successful among these due to its support to local public sector bargaining power against private enterprises, as concluded by Savitch and Kantor [43], such an intervention is likely to be inefficient in the conditions of global digital economy. The same holds with institutionally oriented internationalization of city governments' activities [44], or schemes that highlight some single integrative theme, such as sustainability, livability, inclusion, or community well-being (e.g., [45–47]). All of these are likely to fall short unless they are able to identify and utilize the mechanisms that connect the power of places with the space of flows [16,19].

It looks that the essential element that must be taken into account beside previously discussed globality is digitality. Cities have started to realize that they need to use digital technology to capitalize on the rapidly growing digitally driven economy, which can be oriented toward strengthening urban economy through improved productivity for local businesses [48], by global innovation networking [49] or by adopting platform model as in building local innovation platforms and facilitating local platform players [50,51]. The city governments face a novel challenge associated with platform urbanism as an instance of platform capitalism [52]. Urban communities need sufficient infrastructure to facilitate operations of the platform economy and ability to negotiate with private platform companies and local service providers. They obviously benefit from the economic structure in which the sectors relevant for the platform economy have strong presence. Moreover, localities need talent that is essential for the entrepreneurial, R&D, managerial, and professional functions of the platform economy. Such aspects direct our attention to infrastructures, industries, and talent, which are essential elements of the localization of the activities of the platform economy and, inversely, of connecting the locality to the global platform economy.

## 4. The Case of Helsinki Metropolitan Area

In the following case study, we will discuss local responses to the rise of global platform economy through the experiences of the cities of Helsinki, Espoo and Vantaa in the capital region of Finland (we exclude only a small municipality of Kauniainen from the officially defined capital region). Capital region, also referred to as Helsinki Metropolitan Area (HMA), is a part of a larger Greater Helsinki Region, which is the largest urbanized area in Finland, including the capital region and 10 municipalities surrounding it. The capital region has almost 1.2 million inhabitants, while the population of the Greater Helsinki Region is about 1.5 million. The Greater Helsinki Region is again part of the formal regional structure known as Helsinki-Uusimaa Region, which includes 26 municipalities in all, with a population around 1.7 million, making almost 1/3 of country's total population. The region is an unchallenged center of economy, culture, and competence as well as the most densely populated, diverse, and internationalized area in the country. It accounts for almost 39% of Finnish national GDP [53].

The three cities discussed here are among the largest in Finland, each having their own profile. The city of Helsinki with some 655,000 inhabitants (1 January 2020) is the largest city of the country and the capital of Finland. The population of the city of Espoo is in the region of 290,000 (1 January 2020). The city has fairly dispersed urban structure. It is well known for the main campus of Aalto University in Otaniemi and headquarters in the high-rise business district in Keilaniemi. Lastly, the city of Vantaa has some 233,000 inhabitants (1 January 2020). It is best known for being the host city of Helsinki-Vantaa Airport as well as for multipolar urban structure and a high profile in residential function and multiculturality. These three cities form the country's most important metropolitan area and the largest concentration of high-tech, business services, universities, and cultural institutions.

Its municipalities have increased collaboration in innovation and business development for about twenty years, which has produced, among others, the innovation strategy designed for the whole metropolitan area [54], the establishment of the international trade and investment promotion agency Helsinki Business Hub to serve the entire capital region, and a host of collaborative projects that promote directly or indirectly business development in the metropolitan area [49].

Regarding the factors that are relevant for platform economy, innovativeness, technological advancements, start-up scene, and talent attraction are of great importance. The HMA has some strengths in all such areas. It is the largest and most powerful concentration of population and economic activity in Finland. While being rather small metropolitan area by European standards, it has been considered as one of the leading knowledge-intensive hubs even in global comparison, as exemplified by various international rankings and benchmarking studies [36,55]. For example, the Helsinki-Uusimaa Region is the most innovative region in the EU according to the European Commission's Regional Innovation Scoreboard of 2019. Helsinki ranked eighth in the IMD Smart City Index 2019 among 102 reviewed cities. Helsinki was the third best city in the world for startup companies in 2019 according to Valuer.ai's ranking. Similarly, Helsinki ranks fourth among rising start-up ecosystems in the Global Startup Ecosystem Report (GSER) 2020 produced by Startup Genome and Global Entrepreneurship Network. In the GSER 2019, the gaming industry and related expertise were viewed as one of Helsinki's special strengths given the size of its operational environment. This is epitomized by the rise of such companies as Supercell, the creator of the Clash of Clans, and Rovio, the company behind Angry Birds. [36,56]. These kinds of indicators imply that the capital city of Finland with its neighbors has obvious strengths that are of critical importance in creating connections with the platform economy.

## 5. Empirical Analysis

### 5.1. Understanding Platform Economy

Key professionals and experts involved in local economic development in Helsinki Metropolitan Area have been able to identify a range of challenges associated with the platform economy, yet the conceptualization of this particular aspect of the economy remains rather vague. While the representatives of cities are aware of the phenomenon in general, and are familiar with its various manifestations, there is obvious lack of understanding of its nuances, potentials, and implications for local economic development.

"Is it a familiar concept and is its content clear? Technically speaking, I can say straightforwardly that maybe not to me." (Interview 3)

"It's both a threat and an opportunity; we're not quite sure which one it is at this point." (Interview 5)

"Indeed, it is probably the kind that everyone understands some aspect of it, but all that it will affect–that large-scale change or turning point–well, I think, all that remains pretty unclear to everyone." (Interview 7)

The key issue seems to be the increased ambiguity, for in the industrial era and even during the gradual changes of the 1980s and 1990s epitomized by the rise of technology parks and later by the Internet explosion, the challenge itself appeared more conceivable than the current economic transformation. This was expressed by an interviewee as follows.

"These are just those problems that we . . . cannot figure out even conceptually, that is, what is now the challenge to cities with this rising platform economy. That is what we have here [considered]. Indeed, the 'previous round' was in a sense clearer, we could say, [requiring] just a kind of strategic positioning." (Interview 6)

### 5.2. Local Embeddedness and Disembedding Tendencies of the Platform Economy

Informants pay attention to the inherent ambiguity associated with the localization of the platform economy and the difficulty to instill its forms and logic in local economic development. It seems to be commonly held that digital business environments and global platforms are largely outside the scope of conventional economic development policy. The challenge is thus to learn to utilize platforms as distribution channels locally.

> "For the platform builder it [i.e., the platform] is a business as itself, but for others, it's not really that per se, but it's the basis on which to build a part of the value chain or an aspect of a process. [ . . . ] In my view, it's a distribution channel among many." (Interview 3)

> "The way the platform economy, or a business model like Amazon or Uber or something, is defined globally, that kind of definition of the platform economy does not quite work when applied to the city." (Interview 6)

The platform economy seems to require that development activities are not too tightly bound to direct local interest and local actorship. Rather, platforms and value creation on them requires certain degree of openness to the involvement of nonlocal actors who reap their own share of the values generated within platform or network constellations and business ecosystems. Currently, especially Helsinki and Espoo view the role of globally attractive local digital and hybrid platforms essential for their economic development. However, the idea of harnessing the logic and models of the platform economy per se is still fuzzy in the given context, constrained by city governments' role as democratically governed public entities and by a range of privacy, data security, and other policy issues.

> "In the platform economy, I believe, the most difficult thing to understand will be that those companies don't have to come from Espoo." (Interview 2)

> "Are we also able to build interesting, in a sense place-bound yet digital development platforms and environments that would be linked to one another in the spirit of platform economy because these companies–big or small–would need one another? [ . . . ] Also, for large players in an industry, it's important to know that if they supply certain technology solutions, how and by which means they can build that ecosystem locally, too." (Interview 4)

> "If we take the narrow definition of the digital platform, in which someone opens to products and end-users sort of opportunity to do transactions among themselves and then takes only its share from governing the platform, in this sense the city has very few developed digital platforms that we could use in the same way. And that is a particular kind of restriction in this respect. Besides, there will be to some extent these data security and protection practices regarding the city governments ability to secure its service provision." (Interview 6)

### 5.3. Strategic Approach to the Platform Economy

In city strategies and local economic development strategies, platform economy is not emphasized due to its novelty and ambiguities. Our key informants acknowledge this and the obvious lack of clarity of the transformation caused by platformization, yet they emphasize that there are some elements in their approaches to economic development, such as the idea of enabling local government, that reflect certain aspects of this economic transformation.

> "Neither uses the word 'platform economy' [interviewee refers to the city strategy and the economic policy priorities of the city], [ . . . ] but then again, our entire city strategy starts from the fact that we wish to enable things, and that is strongly associated with platform economy. [The fact] that it is there, is not mentioned explicitly, yet the entire mindset is based on enabling; so if we would be restricting all the time everything and regulating all the time everything, then there would be no chances for platform economy. And this same [thinking] cuts across the priorities of our economic development policy as well." (Interview 7)

"It may be that we do not have a sufficiently strategic grasp of this. [ ... ] In my view, there is no clear integration of the kind of mechanism that would allow us more strategically consider these different technologies, different trends as for example this platform economy, regarding how certain industries are changing, how the consumer behavior and its channels are changing, and how we react to that." (Interview 5)

Cities' responses to the challenges of platform economy vary depending on their conditions, competitive advantages, and economic profiles. For example, the city of Vantaa as a logistics hub has not explicitly addressed the opportunities of platform economy simply because such economic players locate primarily in Helsinki and Espoo.

"We pursue a strategy according to which we don't focus on them at all [interviewee refers to platform companies and content providers, such as gaming companies referred to by an interviewer]. In other words, it is, at times, somewhat difficult with differences in the industrial structure, as we know that the gaming companies don't really want to come here. They rather cluster in Helsinki, and when thinking of software more broadly as well as other ICTs, then in Espoo, too." (Interview 5)

All the cities recognize the importance of tracking trends associated with the platform economy. The approach is primarily incremental in the sense that the aspects of platform economy are integrated into existing clusters and industries rather than aiming at radical breakthroughs by chasing major platform companies. Their approach reflects in this sense the premises of the endogenous growth theory.

"We've not necessarily been particularly actively involved in platforms, but it might make perfect sense to take a more active role in this." (Interview 5)

"We're not aiming here at 'double-click world,' some new institution or important actor, but rather, bringing them inside the clusters, industries, I think, is how we might succeed." (Interview 3)

Representatives of city governments and development agencies in the capital region are cautious about city government's ability to meet the challenges of the platform economy alone. Some informants emphasize the role of intercity collaboration, such as the Six City Strategy within which six largest Finnish cities—Helsinki, Espoo, Vantaa, Tampere, Turku, and Oulu—share their experiences about developing smart, sustainable and human-centric platforms and applications (6AIKA in Finnish, see at https://6aika.fi/en/frontpage/). Some informants see that the challenges of platform economy require collaboration between cities and the state as well.

"Let's take the Six City Strategy, the six largest cities in Finland, with varying themes. The greatest achievement of a small country, maybe also in a certain sense a learning outcome, has been how the six cities have first learnt to trust one another, [ ... ] the only advantage of this kind of small nation is the asset generated by working together." (Interview 3)

"I believe that cities may be too small. Rather, we should have cities and the national level pondering together how it would be possible for Finns to do business and build more firmly such a platform." (Interview 5)

*5.4. Nascent Policy Measures and Actions*

The representatives of cities shared the view that chasing global platform companies would be a waste of resources. Rather, actions are directed towards creating enabling environments, facilitative structures, and locally-rooted platforms that connect local actors and their operations with the platform economy, on the one hand, and working on piecemeal improvements and taking actions in which selected aspects of the platform economy have been taken into account, on the other (Interview 5). The building of digital and hybrid urban platforms and living labs have been the first steps in learning about how the platform economy relates to the locality and its processes.

"In Espoo, we might have already done some things associated with the platform economy, without really realizing it, for we've been opening the city as a kind of platform. Just like the service center in Iso Omena [shopping mall] is a really good example of how we've already done it, without paying attention to it. The issue is how we could adopt a way of thinking that aligns more with platform economy." (Interview 2)

While the cities in small open economies like Finland are not usually able to attract headquarters of platform companies or other major players of the platform economy, they can offer an attractive environment to application developers, content providers, and back-office service providers. The capital region is a vibrant environment in this respect, and it produces a lot of ideas, experiments, and applications that reflect the platform logic, of which our informants mentioned DriveNow car sharing service, School-as-a-Service concept, Uber of the Seas, virtual home care, and a host of other recently launched solutions (Interviews 2; 4; 6). They are early instances of platform urbanism in which local governments have a role to play as enablers, facilitators, and regulators.

"It seems that the platform builders always take their own cut, but when one builds on it, there is another cut, of which gaming industry is evidently a good example. But there must be a host of other similar applications and solutions on which it is possible to build a range of different kinds of solutions." (Interview 5)

"In Helsinki, now [2018] they pilot a kind of Uber of the Seas and hope that it will happen, that is, in practice it deals with access to archipelagos. It is in our interest that people can get there, to enable recreational use. It is not economically viable to open ferryboat connection, but it is sensible that we designate stops and that there are in a way structures in place, but then the transaction takes place through a platform, thus someone selling a boat [ride] and another paying for the boat time." (Interview 6)

Practically all informants mentioned the opening of data as one of the most important areas in which cities in the capital region have put some efforts in order to support urban innovations and generate businesses (Interviews 1; 2; 3; and 6). A prime example of such efforts is Helsinki Region Infoshare, which was mentioned in passing in several interviews.

"Our goal has been to generate, through open data, opportunities for others to create business." (Interview 6)

"I believe one of the most concrete platform economy initiatives has been the opening of data." (Interview 3)

The features of the platform economy and related economic trends point to the relevance of innovativeness and further to capabilities and talent, which was recognized by many of our informants. Regarding local connections with the new economy, attention in urban development is not anymore directed primarily to traditional infrastructure issues but to knowledge infrastructure, inspiring start-up scene, and talent creation, attraction and retention.

"Well, how to attract skilled people to the city is probably the most important issue in the end." (Interview 6)

Lastly, for cities, the challenge of dealing with the platform economy is to a large extent that many of the preconditions and operations of digital economies remain outside the reach of local authorities and local development agencies. Cities have to understand the logic of sharing and platform economies, which are primarily run by private-sector actors with the help of digital platforms. In many cases, even big platform players need connections with the local business and service ecosystem, which creates opportunities for local economic development. The show is not run by the city but by a host of players with different roles in the platform economy. This view takes discussion towards value networks and ecosystem thinking. Eventually, it brings local public services and the quality of life issues into the picture.

"So, do we have any specific, exciting expertise to offer internationally for the creation of platform economy and its preconditions? When companies are searching for investment opportunities and location, at least among our client companies always the first reason why they come to Helsinki [is] know-how, [then] the exciting start-up buzz, and the third reason may vary; some say it may be the services we offer, or otherwise it may be something like a safe environment, or good educational opportunities for children." (Interview 4)

## 6. Discussion

Empirical analysis shows that the metamorphosis from the growth machine thesis towards platform urbanism is far from unambiguous. Anomalies in the local economic development appeared as the current realities do not seem to match with the view of localities as growth machines that reflect the interests of the owners of land and immovable properties. While such local interests have maintained their relevance in some respects, and they are taken into account by local authorities in the design of local development policy, the financialization of the economy, growing importance of knowledge-intensive portion of global flows, and the impact of digital transformation on virtually all aspects of the economy [34], have become increasingly important factors in the local economy, implying that transnational capitalist class has ever greater impact on it. It is caught between location specificity and translocal accumulation tendencies, the latter assumed to be a stronger element in the strategic decisions made by the members of this class. Interestingly, the location becomes assessed primarily in terms of its ability to attract and support productivity, innovativeness, and profitability, which due to creative class tendencies and the conditions of cognitive-cultural capitalism bring few—even if somewhat elitist or instrumental—social considerations into the picture (cf. [57]).

This structural setting reveals the strategic role the democratic city governments play as mediators between localities and global capital [19]. Our results show that cities have difficulties in conceiving the most elusive aspects of the new economy, of which we focus here on the platformization. Cities have started to embrace their role as enablers and facilitators and created urban platforms, which reflect platform thinking but form a dubious basis for understanding the relationship between the city and the platform economy. The difficulties of the representatives of city governments and local development agencies to grasp the local-global nexus in terms of the platform economy reflects the fact that this relationship is inherently complex or fuzzy.

The difficulties in understanding and conceptualizing the logic of the platform economy matter simply because platforms are disruptive, and they eventually reorganize essential aspects of the economy. A critical strategic issue in this respect is the ability to create or maintain local embeddedness of operations that are connected with the globally oriented platform economy. Cities in the Helsinki Metropolitan Area have vague ideas of what this means but have primarily relied on locally-rooted innovation platform model, which has its limitations in terms of making the city as a smart, proactive and strategic player on this scene. That said, the cities of Helsinki and Espoo, and to a lesser extent Vantaa, have already adopted new approaches and strategies that reflect this change, as seen in their attempts to create enabling environment for local development, set up urban development and innovation platforms, and utilize business and innovation ecosystems, as expressed in their strategy documents. Even if the terminology is not applied in a consistent manner in the key policy documents, especially the cities of Helsinki and Espoo have paid attention to platform and ecosystem thinking and the enabling role of platforms in the promotion of local development (see [58,59]). There is a brief reference to city as a platform also in the city strategy of Vantaa [60]. Thus, there are already signs of shifting towards platform-oriented approach. Most notably, Helsinki as the most cosmopolitan city in Finland, has started to apply platform thinking and innovation-driven economic development, as formulated in the Helsinki City Strategy 2017–2021. Such thinking is epitomized in the operations of Forum Virium Helsinki, which is a dynamic innovation company of the city, aiming at putting the ideas of culture of experimentation and user-driven innovation into practice [58]. In the same vain, Espoo as a high-tech hot spot, has started to deploy ecosystem and platform thinking in its business

promotion and service development [61,62]. Espoo's city strategy crystallizes such a strategic goal as follows: "The promotion of internationally appealing business and experiment platforms that diversify the economic structure of Espoo is at the core of the city's growth strategy." [59].

The three cities in the capital region have adopted rather wobbly policies that respond to or attempt to utilize the emerging platform economy. In general, the understanding of the platform urbanism has not matured yet (see e.g., [63]). It is perceived from a somewhat localist perspective, even if various aspects of local-global dialectic have already found their expressions in city strategies. The emergence of locally embedded digital and hybrid platforms is exemplified by such cases as Solution Factory in Vantaa, School Community Living Lab in Espoo, and Smart Kalasatama in Helsinki ([51]; see also https://citybusiness.fi/en/platforms/). The current approach to the challenges of local-global dialectic is manifest in the role played by local innovation platforms with their connections with wider knowledge infrastructure of the region, which as the major locally rooted resource integrators and facilitators epitomize the current state of development of local adjustment to global digital economy. The other area in which cities have collaborated proactively revolves around open data that feeds new economy as a collaborative effort within HRI and Six City Strategy. The third element that was recognized in the interviews was know-how or talent, which has been on the regional development agenda for some time. Helsinki Business Hub as the international trade and investment promotion agency for the Finnish capital region has started to pilot talent attraction model that aims at alleviating the technology talent shortage in the region [64]. It provides access to innovation platforms, international investors, piloting opportunities, matchmaking, and soft landing services, which indicate that various aspects of talent attraction have become vital part of the promotion of urban and regional economic development.

We may encapsulate such efforts by saying that the efforts of the cities of the capital region associated with the platform economy are visible in the establishment of local innovation platforms as a new form of enhancing collaborative and innovation-driven activities; providing and utilizing open data that fuels the new economy; and attracting talent using talent attraction programs, innovative milieu and thriving start-up scene in order to entice the people who are able to create the greatest value for the regional economy.

## 7. Conclusions

This article has shed light on how the shift towards the global digital economy condition urban economic development. In order to illustrate this transition, we discussed how the growth machine thesis that used the land as the vital urban nexus in the context of industrialization, seems to be less and less accurate in theorizing the current preconditions for urban economic development. As a paradigmatic aspect of the change in the economy, we discussed platform economy, in which the transnational capitalist class operates through digital platforms. In historical perspective, place-based growth machine indeed seems to be transforming into globally connected urban growth platform. Due to such a transformation, a place as a central commodity and related land-use intensification are becoming subordinate to extralocal dimensions of value cocreation within business ecosystems that reflect the condition of global digital economy. This hints that "the city as innovation machine" [65] is affected by the interplay of digital platform logic and the globalized circuits of the accumulation of capital, which generates novel features that go beyond localization and urbanization economies, thus reinforcing tendencies that reflect the interests of transnational capitalist class [66]. However, rather than displacing growth machine altogether, platformization seems to radicalize both ends of the local-global dialectic, as suggested in the theorizations of platform urbanism. As the local side of this equation—including economic development policy—is in democratic postindustrial societies primarily channeled through local democratic institutions, our empirical analysis sheds light on this issue from the point of view of local government.

Let us turn to the findings of our empirical analysis of the three major cities of the Finnish capital region, those of Helsinki, Espoo, and Vantaa. First, our analysis reveals that the representatives of local economic development units have not been able to fully comprehend the complex phenomenon

known as platform economy nor have they been able to design policies that match its scale and complexity. Second, the representatives of cities have conceived the overall challenge of utilizing local embeddedness while at the same time facilitating connections with global markets and business ecosystems that contain extralocal elements. Despite certain degree of ambiguity of such a challenge, the approach of key players responsible for local economic development has started to lean towards platform and service ecosystem thinking, which paves the way for breaking away from narrow-minded localism. Third, strategic documents of all three cities, most notably those of Helsinki and Espoo, reflect such an orientation, depicting the idea of city-as-a-platform [51]. Lastly, city governments' efforts to adjust to global digital economy are visible in the establishment of local innovation platforms and living labs, increased interest in startup and talent attraction and the establishment of talent attraction programs, and as a special action to fuel data-driven economy, making public data freely available through open data service.

We have discussed in this article local adjustment to the global digital economy by providing empirical evidence of how the key actors of the three cities of the HMA conceive the platform economy and what policy measures these cities have designed to meet the challenges of such a contextual change. We are dealing with a phenomenon that is both complex and novel. The demand for further research on this topic is obvious, for each aspect of this setting requires elaboration and sophisticated theoretical and empirical analyses. From the point of view of cities, a major challenge is to provide scientific knowledge of various aspects of platform urbanism that manifests local-global dialectic. Getting a sharper view of how localities are connected with the platform economy and how such a condition should be taken into account in local economic development policy is a critical task as cities strive to cope with changing economic realities.

**Author Contributions:** Conceptualization, A.-V.A., M.L., and H.L.; methodology, A.-V.A., M.L., and H.L.; formal analysis, A.-V.A., M.L., and H.L.; writing—original draft preparation, A.-V.A.; writing—review and editing, A.-V.A., M.L., and H.L. All authors have read and agreed to the published version of the manuscript.

**Funding:** This research received no external funding.

**Acknowledgments:** We wish to express our thanks to Johanna Tiitinen for providing the first draft of the translations of the selected verbatim quotations from Finnish into English.

**Conflicts of Interest:** The authors declare no conflict of interest.

## Appendix A

### Interviews

Harri Paananen, Economic Development Manager, city of Espoo, interviewed by M.L. and H.L. on 15 May 2018.
Tomas Lehtinen, Data Analytics Consultant, city of Espoo, interviewed by H.L. on 6 June 2018.
Jose Valanta, Director of Economic Development, city of Vantaa, interviewed by M.L. and H.L. on 6 June 2018.
Marja-Liisa Niinikoski, CEO, Helsinki Business Hub, Helsinki, interviewed by M.L. and H.L. on 6 June 2018.
Kimmo Viljamaa, Economic Development Manager, city of Vantaa, interviewed by M.L. on 11 June 2018.
Santtu von Bruun*, Head of Competitiveness and International Affairs, city of Helsinki, interviewed by A.-V.A. and H.L. on 15 August 2018.
Marja-Leena Rinkineva*, Director of Economic Development, city of Helsinki, interviewed by A.-V.A. and H.L. on 15 August 2018.
* von Bruun and Rinkineva were interviewed at the same time (joint interview).

## Appendix B

**Interview questions**

1. **Understanding platform economy**

   - How the representatives of city government and local development agencies perceive and conceptualize the platform economy or more broadly global digital economy?

     a. Is the platform economy clear or unclear as a concept?
     b. How it can be seen in city strategy or economic development strategy?
     c. How does it affect orientation or the selection of policy lines in city government?
     d. How do global platforms and new business models—Facebook, YouTube, Amazon, Airbnb, Uber—affect the life in the city? How do they affect public administration, events, local business, services, etc.?

2. **Local interests and responses regarding the adaptation to the platform economy**

   - How have cities conceived local interests and local embeddedness of economic development in the context of the globalized digital platform economy?

     a. How the cities secure that local interests are taken into account in the operations associated with global digital economy?
     b. Implications for the development of industries, clusters, and economic sectors?
     c. Regarding platform economy, how the local development relates to institutional landscape and different groups of actors: global platform companies; international and European business; IT sector, game industry etc.; and local players of the platform economy (e.g., content providers), educational and research institutions, and relevant local projects, such as Helsinki Region Infoshare (HRI).

3. **City strategies and economic development policies as a precondition for building a success in the platform economy**

   - How have cities addressed the aspects of the platform economy in their strategies and in their economic development policies?

     a. What is the major emphasis in the city strategy regarding the promotion of economic growth?
     b. What are the spearheads of economic development policy and how the platform economy relates to them?
     c. How platform economy is seen in relevant city strategies and economic development policy documents?

4. **Concrete policy measures and actions**

   - What are the factual policy measures designed and actions taken to meet various challenges associated with the platform economy?

     a. Special policy measures, such as urban design, industrial and cluster policies, and talent attraction?
     b. The role of IT firms, platforms, sharing economy, content production, cloud computing services and other platform players?
     c. What is the role of the projects of cities or regional authorities?
     d. The role of universities and research institutes?
     e. How does the development of platform economy relate to the development of urban structure or physical urban environment?
     f. The relevance of talent in the platform economy: how are experts in the IT field attracted to the city?

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
