# Peer review of "City as a Growth Platform: Responses of the Cities of Helsinki Metropolitan Area to Global Digital Economy"

_urbansci, doi:10.3390/urbansci4040067_

Round 1

Reviewer 1 Report

Dear authors,

I congratulate for your idea to study structural changes towards the global digital economy in the urban economic development.

I have the following suggestions:

  1. You should highlight the research objectives in the Introduction section. 
  2. The research methodology must be improved. Even though you presented arguments that led you to this method, you did not specified how did you choose those persons as respondents (see the next item), you did not explain how did you get to the interview guide and how did you connect the questions with your objectives. 
  3. Even though you presented (Appendix 1) the name and the organizations that the interviewed persons are representing, we do not understand how did you choose those persons. I suggest to specify (maybe in a table) the expected outcomes from each respondent. I also suggest to present the interview guide (another appendix)
  4. I do not agree with the structure of Section 5. It has to be splited in 2 parts: the first part is about Helsinki Metropolitan Area and the second is about the themes of research and how did the respondents answer.
  5. In 5.4. subsection, lines 561-568, you mixed the answers with some issues from official documents. I suggest that you should do that in the Discussion section. It is, also, happening in other subsections.
  6. The Conclusions section is not consistent and it does not give enough arguments for the writing the article. The conclusions should be structured following the results and they should be related to the actual state of knowledge.

Author Response

We thank reviewer 1 for providing constructive comments on our paper, which we have taken into account in our revision.

  • We have transferred discussion about research objectives into introductory section.
  • We have improved methodology by specifying on what grounds we chose our informants and attaching the structure of our thematic interview in Appendix B.
  • We have discussed the selection of interviewees in the methodology section. They represent the highest level managers of economic development units of three cities. Expected outcomes can be traced from our research questions. Appendix B serves and an “interview guide”. We follow conventional semi-structured interview scheme.
  • We separated the case description from empirical analysis, as suggested.
  • We removed references to strategy documents from the sections that discuss the results of the interviews, and placed them to discussion section, as suggested.
  • We tried to sharpen concluding section as suggested by the reviewer.

Reviewer 2 Report

This is an interesting study to help understand the development of cities under the global economic system.

In particular, the research problem setting is interesting and the theoretical background is systematic. Also interesting is the content and interpretation of interviews with people in key positions in the city.

However, the main data of the study relies on interviews.

Basic information is provided on the city of interest, but very little information directly related to this study, such as development, innovation, digitization of the region, is allocated (5.1 The case of Helsinki Metropolitan Area). The content related to this should be supplemented even briefly.

Besides, although the qualitative research method is used in this study, an expression appears as an empirical study used in a quantitative approach. It needs to be revised.

Author Response

We thank reviewer 2 for providing constructive comments on our paper, which we have taken into account in the revision. As suggested by the reviewer, we have included a description of HMA’s features in terms of development, innovation and digitization, as requested. We believe that this improved the section in question. While doing so, we also added a few relevant references.

This is an empirical analysis, in which we have relied in case study methodology, and within that framework, utilized primarily semi-structured interviews. The current version reflects such methodological underpinnings. We have also added a few details concerning informant selection and the structure of the interview questions in Appendix B.

Reviewer 3 Report

The relationship between the new platform economy and urban policy is the latest research theme, and it can be usually evaluated that the trends are summarized in the first half. The point of this paper is to try to add more solid movements and new findings through interviews in the Helsinki metropolitan area of Finland. However, it is hard to say that there is sufficient proof on that point. Regarding this point, I feel that by considering the content of the interviews a little more, the direction of the city policies that are currently going on can be suggested. However, it can be judged that there is some original significance in showing the timeliness of this theme and the trends of new urban policies.

Author Response

We thank reviewer 3 for providing constructive comments on our paper, which we have taken into account in the revision. Concerning the ‘proof’ of our point, it is true that it difficult to provide conclusive empirical evidence of such a complex and novel phenomenon. Yet, we hope this contribution is a one step further in such a challenging academic endeavor. On the basis of the reviewer’s comments, we have added a few new passages to our empirical analysis, which hopefully brings relevant additional content to our analysis. Due to the novelty and ambiguities of the phenomenon in question, informants were not able to point clear future directions. Besides, this particular aspect was not included in the main themes of our semi-structured interview. This is an important area that should be elaborated in the future within some other research project. As pointed out by the reviewer, our main emphasis was on the description of the current state of development.

Round 2

Reviewer 1 Report

I noticed that you considered all my suggestions. Again, I appreciate your work and the researched theme.

Best regards!